# Derivation and validation of an algorithm to predict transitions from community to residential long-term care among persons with dementia—A retrospective cohort study

**Wenshan Li**[1]*, **Luke Turcotte**[2], **Amy T. Hsu**[3,4], **Robert Talarico**[5], **Danial Qureshi**[6], **Colleen Webber**[1,3], **Steven Hawken**[1,3,7], **Peter Tanuseputro**[8], **Douglas G. Manuel**[1,5,9], **Greg Huyer**[10]

1 Ottawa Hospital Research Institute, Ottawa, Ontario, 2 Broke University, Department of Health Sciences, St. Catherines, Ontario, Canada, 3 Bruyère Research Institute, Ottawa, Canada, 4 University of Ottawa, Brain and Mind Institute, Ottawa, Canada, 5 ICES uOttawa, Ottawa, Canada, 6 University of Oxford, Nuffield Department of Population Health, Oxford, United Kingdom, 7 University of Ottawa, School of Epidemiology and Public Health, Ottawa, Canada, 8 University of Hong Kong, Department of Family Medicine and Primary Care, Hong Kong, 9 University of Ottawa, Department of Family Medicine, Ottawa, Canada, 10 Health Canada, Ottawa, Canada

* wensli@ohri.ca

**Data Availability Statement:** Data used for this study is held securely in encoded form at ICES. While data sharing agreements prohibit ICES from

## Abstract

### Objectives

To develop and validate a model to predict time-to-LTC admissions among individuals with dementia.

### Design

Population-based retrospective cohort study using health administrative data.

### Setting and participants

Community-dwelling older adults (65+) in Ontario living with dementia and assessed with the Resident Assessment Instrument for Home Care (RAI-HC) between April 1, 2010 and March 31, 2017.

### Methods

Individuals in the derivation cohort (n = 95,813; assessed before March 31, 2015) were followed for up to 360 days after the index RAI-HC assessment for admission into LTC. We used a multivariable Fine Gray sub-distribution hazard model to predict the cumulative incidence of LTC entry while accounting for all-cause mortality as a competing risk. The model was validated in 34,038 older adults with dementia with an index RAI-HC assessment between April 1, 2015 and March 31, 2017.

making the dataset publicly available, access may be granted to those who meet pre-specified criteria for confidential access (www.ices.on.ca/DAS). The data as well as codes for cohort creation and analyses are available from ICES upon request (contact: das@ices.on.ca), with the understanding that certain programs may rely upon coding templates or macros that are unique to ICES and are therefore either not fully accessible or may require modification. Parts of this material are based on data and/or information compiled and provided by CIH, Ontario Health and the Ontario Ministry of Health. The analyses, conclusions, opinions and statements expressed herein are solely those of the authors and do not reflect those of the funding or data sources; no endorsement is intended or should be inferred.

**Funding:** This study was funded by the Canadian Institutes of Health Research (CIHR), grant No. PJT-173346. This study was supported by ICES, which is funded by an annual grant from the Ontario Ministry of Health and the Ministry of Long-Term Care. WL is supported by CIHR and Associated Medical Services (AMS) postdoctoral fellowships. The analyses, conclusions, and statements expressed herein are solely those of the authors and do not reflect those of the funding or data sources; no endorsement is intended or should be inferred.

**Competing interests:** The authors have declared that no competing interests exist.

## Results

Within one year of a RAI-HC assessment, 35,513 (37.1%) individuals in the derivation cohort and 10,735 (31.5%) in the validation cohort entered LTC. Our algorithm was well-calibrated ($E_{max} = 0.119$, $ICI_{avg} = 0.057$) and achieved a c-statistic of 0.707 (95% confidence interval: 0.703–0.712) in the validation cohort.

## Conclusions and implications

We developed an algorithm to predict time to LTC entry among individuals living with dementia. This tool can inform care planning for individuals with dementia and their family caregivers.

### Author summary

We developed and tested a new model to predict how long before people with dementia will be admitted to long-term care (LTC) facilities. Using routinely collected data on all older adults living in Ontario receiving home care between April 2010 and March 2017, we created our prediction model and tested its performance using various metrics. We found that, within a year of their initial assessment, about 35% of persons with dementia were admitted to LTC. We also determined that our model performed well. This model can help healthcare providers and families better plan for future care needs by providing a clearer picture of when someone with dementia might need to move to a long-term care facility.

## Introduction

In Canada, 61% of older adults living with dementia reside in their own homes [1]. Access to publicly-funded home care programs–designed to supplement care provided by family and friend caregivers (henceforth referred to as "caregivers")–generally provide a safe and cost-effective alternative to residential long-term care (LTC; i.e., publicly-funded institutional facilities that provide 24-hour nursing and personal care support; also known as nursing homes) [2]. However, many caregivers experience heavy burden, committing long hours (26 hours per week on average in Canada [1]), performing physically-demanding and stressful tasks [3–5], and incurring significant out-of-pocket costs [1]. As a result, caregivers of persons with dementia are twice as likely as caregivers of persons without dementia to be distressed [1], leading to earlier institutionalization of their care-recipients as their cognitive and functional impairments increase [6]. Thus, dementia remains a key contributor of LTC placement [7]; within 5 years of dementia diagnosis, nearly 50% are admitted to a LTC[8], and up to 71% of newly admitted LTC residents in Ontario have a dementia diagnosis [9]. Thus, individuals with dementia and their caregivers may benefit greatly from tools that can estimate future care needs.

The placement process for LTC in Canada involves a multi-part assessment that starts with an assessment by a Home and Community Care Support Services care coordinator to review the individual's health, behaviour, and care needs [10,11]. In several Canadian provinces, including Ontario, the interRAI multidimensional health assessments are used to assess needs for publicly-funded home care and to determine eligibility for LTC admission [12]. Several

clinical decision-support algorithms can be calculated using information from these assessments; including the Method for Assigning Priority Levels (MAPLe) algorithm [13], which stratifies home care recipients into five priority levels and is used to systematically prioritize home care clients for admission to LTC [13,14]. While MAPLe can differentiate home care clients when predicting time to LTC placement[13], its predictive capacity among individuals with dementia is limited since the majority of this sub-group are classified into the highest MAPLe categories [15]. Furthermore, MAPLe was developed primarily for use by clinicians and care coordinators and have not been used by clients or caregivers.

Currently, there exists no prognostic tools, to our awareness, for individuals with dementia to estimate time between current health state and the need for residential care. Therefore, we aimed to develop a multivariable regression model to estimate time-to-LTC admission after dementia diagnosis in community-dwelling older adults who were receiving home care. This model focuses on self-reportable questions and will form the basis of a risk communication tool that can be used by the general public to make informed decisions about future care needs.

## Methods

### Data sources

Using uniquely encoded identifiers, we linked and analyzed multiple health administrative databases held at ICES (formerly known as the Institute for Clinical Evaluative Sciences), an independent, non-profit research institute funded by the Ontario Ministry of Health and Long-term Care. As a prescribed entity under Ontario's privacy legislation, ICES has been authorized by the Information and Privacy Commissioner of Ontario to collect and use health care data for the purposes of health system analysis, evaluation and decision support. Secure access to these data is governed by policies and procedures that are approved by the Information and Privacy Commissioner of Ontario. Projects, such as this study, that use only data collected by ICES under section 45 of PHIPA are exempt from Research Ethics Board review and does not require consent to participate. All methods were carried out in accordance with relevant guidelines and regulations.

The primary source for our predictors is the Resident Assessment Instrument–Home Care (RAI-HC), which is completed for all individuals in Ontario in need of home care or LTC. The RAI-HC is provided to all individuals applying for LTC placement. Additionally, it is used to routinely assess (every six months or when a change of status occurs) individuals receiving publicly-funded home care. The RAI-HC contains information on individuals' characteristics, functional and health profiles, and service use [12,16]. The inter-rater reliability of items on the RAI-HC [16] is high and the disease diagnoses items have been validated [17]. We linked individuals' RAI-HC assessments to additional datasets: 1) the Registered Persons Database, which captures demographic information including sex, age, and dates of birth and death; 2) the Local Health Integration Network (LHIN) Database which provides geographic information on LTC homes; and 3) the Ontario Health Insurance Plan (OHIP) and Ontario Drug Benefits (ODB) claims database, used in the algorithm to capture LTC admissions.

### Study design and cohort

We conducted a retrospective cohort study of all community-dwelling older adults diagnosed with dementia and subsequently assessed with the RAI-HC between April 1, 2010 and March 31, 2017. Individuals were over 65 years of age on the date of their first RAI-HC assessment after dementia diagnosis ("index assessment") and had never been a resident in LTC homes. The derivation cohort comprised of 95,813 persons (750,526 person-weeks of follow-up)

whose index assessment occurred prior to March 31, 2015. The validation cohort included 34,038 persons (1,230,289 person-weeks of follow-up) with index assessment between April 1, 2015 and March 31, 2017. Each person was followed for a maximum of 360 days after index assessment, censored at the end of follow-up, date of death, or loss of OHIP eligibility, which-ever occurred first. We followed the Transparent Reporting of a multivariable prediction model for individual Prognosis Or Diagnosis (TRIPOD; S1 Checklist) reporting guideline [18] for the presentation of our methods and results.

## Case ascertainment

Dementia diagnosis was determined using a combination of information from the RAI-HC assessment and a previously validated algorithm (sensitivity of 79.3%, specificity of 99.1% when validated against EMR data) [19]. This algorithm identifies individuals as having been diagnosed with dementia if they meet at least one of the following criteria: 1) one hospitaliza-tion with a diagnosis for Alzheimer's or dementia-related diseases according to the Interna-tional Statistical Classification of Diseases and Related Health Problems, 10th revision; 2) three physician claims codes for dementia-related diseases at least 30 days apart in a two year period; 3) a prescription for medications in the cholinesterase inhibitor sub-classes to treat Alzhei-mer's and dementia-related diseases. Date of dementia diagnosis was defined as the date of the first of the aforementioned hospitalization, physician claims, or prescription. Considering its moderate sensitivity and known under-diagnosis [20,21], we supplemented the algorithm by including individuals with a dementia diagnosis and a Cognitive Performance Scale (CPS) score equal or greater to 2 on their RAI-HC assessment. This supplemented definition has been adopted by the Public Health Agency of Canada (PHAC) and shown internally to increase the number of dementia cases captured by 8%.

## Predictors

We identified 56 potential predictors from the index RAI-HC assessment. The selection of pre-dictors was informed by clinical experience and a review of literature for common risk factors and predictors of nursing home/LTC entry among individuals with dementia. They include socio-demographics (i.e., age, sex, level of education), presence of co-morbidities (e.g., cancer) and behavioural issues (e.g., wandering), availability of caregivers and indicators of caregiver distress, home environment, and several validated composite measures of functionality, mood, and cognition including the CPS [22], Activities of Daily Living Hierarchy Scale (ADL-H) [16], the Instrumental Activities of Daily Living Hierarchy Scale (IADL-H) [16], the Depres-sion Rating Scale (DRS) [23], and the Changes in Health, End-Stage Disease, Signs, and Symp-toms Scale (CHESS) [24]. We also included self-reported use of health care (e.g., number of hospital admissions or emergency department visits in the last 90 days) and home care services (e.g., personal support), medication use, availability of a caregiver and indicators of caregiver distress, receipt of life-sustaining therapies (e.g., dialysis), and symptoms of reduced physio-logic reserve (e.g., weight loss, edema) indicating approach to end-of-life. Additionally, we included cohort characteristics (e.g., type of and reason for assessment) that may account for remaining heterogeneity in the estimated risks. Variables were pre-specified without any step-wise variable selection.

## Outcome

The outcome for our algorithm is time from index RAI-HC assessment to the date of incident admission to LTC. We used a previously validated algorithm which identifies an incident admission into LTC homes if there are two physician claims indicating that the service was

provided in LTC, two prescription drug claims dispensed from LTC, or one physician claim and one prescription drug claim within 30 days of each other from LTC[25]. The sensitivity, specificity, positive predictive value, and negative predictive value for this algorithm were 93.3%, 99.9%, 96.2%, and 99.9%, respectively [25,26]. We used this algorithm since admission records to LTC were not systematically available through the Continuing Care Reporting System during early periods of our cohort capture.

### Statistical analysis

Descriptive statistics were performed to compare individuals who entered LTC, died, or experienced neither outcome within one year of index assessment. A multivariable Fine-Gray [27] sub-distribution hazard regression model was used to predict the cumulative incidence of LTC entry while considering all-cause mortality as a competing risk. All predictors were categorical except for age, which was modelled using restricted cubic splines with 5 knots at the 5th, 27.5th, 50th, 72.5th and 95th percentiles of its distribution. Missingness was considered a separate category in the modelling, with education (45.3%) and time since last hospital stay (28.0%) having the largest proportions of missingness. To achieve parsimony, we removed the following predictors as they were collinear (identified using the SAS VARCLUS procedure [28]) with other predictors: bowel incontinence, bladder incontinence, mood decline, long-term memory recall, and presence of psychiatric diagnosis. Our final model included 51 predictors with 99 degrees of freedom.

Model performance was evaluated on the validation cohort in terms of discrimination (i.e., Harrell's c-statistic) and smoothed calibration (e.g., plotting the observed versus predicted probabilities of LTC entry) [29]. We also generated two calibration statistics: the "$E_{max}$" by Harrell et al., which is a measure of the maximal absolute difference between observed and predicted probabilities of the outcome [30]; and the average Integrated Calibration Index ($ICI_{avg}$), which is similar to $E_{max}$ but weighted by the empirical density function of the predicted probabilities [31]. All calibration curves and indices accounted for the competing risk for death according to methods described by Austin et al. [29]. Finally, model coefficients for algorithm implementation were generated for the total cohort.

We also performed several sensitivity analyses to evaluate the calibrations of our algorithm stratified by the following variables: individuals' ADL, IADL, and CPS scores, and whether the individual was assessed at hospital at index RAI-HC. We further examined regional variations by evaluating event rates and calibration by LHIN. All statistical tests were two-tailed with p<0.05 indicating statistical significance. All analyses were performed using SAS Enterprise version 9.4 [32].

## Results

Select characteristics of persons in the derivation and validation cohorts are presented in Table 1 (see S1 Table for the complete list of characteristics). The overall cohort was mostly female (61.1%), widowed, divorced, or separated (56.4%), and aged 83.7 (SD = 7.1) years on average at assessment. Coronary heart disease (24.0%), stroke (19.0%), and diabetes (18.0%) were among the most prevalent comorbidities. Over 75% of individuals required at least maximal assistance (IADL-H: 5+) to perform instrumental activities of daily living (e.g., housework, meal preparation). Most individuals (63.0%) required at least limited assistance (ADL-H: 2+) for maintaining hygiene, toilet use, locomotion, or eating. Over 30% had moderate to very severe cognitive impairment (CPS: 3+), and 2% to 13% reported various behavioural issues. The derivation and validation cohorts were similar in age, sex, and health and functional status, but more individuals in the derivation cohort were assessed while at hospital (17.5% versus 12.9% of the validation cohort). Finally, the primary caregivers of individuals with dementia

**Table 1. Select characteristics of the total, derivation, and validation cohorts.**

| | | n (%) | | |
|---|---|---|---|---|
| **Characteristics of Individuals with Dementia** | | **Total Cohort n = 129,851** | **Derivation Cohort n = 95,813** | **Validation Cohort n = 34,038** |
| Sex | Female | 79,334 (61.1%) | 58,859 (61.4%) | 20,475 (60.2%) |
| Age | Mean ± SD | 83.7 ± 7.08 | 83.6 ± 7.00 | 84.0 ± 7.31 |
| Marital status | Married | 49,659 (38.2%) | 36,982 (38.6%) | 12,677 (37.2%) |
| | Widowed | 64,546 (49.7%) | 47,727 (49.8%) | 16,819 (49.4%) |
| | Separated or divorced | 8,745 (6.7%) | 6,142 (6.4%) | 2,603 (7.6%) |
| | Never married | 5,452 (4.2%) | 3,930 (4.1%) | 1,522 (4.5%) |
| | Other | 1,449 (1.1%) | 1,032 (1.1%) | 417 (1.2%) |
| Time since last hospital stay | Presently in hospital | 21,207 (16.3%) | 16,803 (17.5%) | 4,404 (12.9%) |
| | Within last 30 days | 21,458 (16.5%) | 14,878 (15.5%) | 6,580 (19.3%) |
| | More than 30 days ago | 50,883 (39.2%) | 37,972 (39.6%) | 12,911 (37.9%) |
| | Missing | 36,303 (28.0%) | 26,160 (27.3%) | 10,143 (29.8%) |
| Person lives with others | | 13,758 (10.6%) | 9,967 (10.4%) | 3,791 (11.1%) |
| **Functional and Health Status** | | | | |
| ADL Self-Performance Hierarchy (ADL-H) | Independent or supervision (score of 0 or 1) | 69,340 (53.4%) | 51,980 (54.2%) | 17,360 (51.0%) |
| | Limited/extensive assistance for some/all activities (score of 2 to 4) | 50,819 (39.1%) | 36,714 (38.3%) | 14,105 (41.4%) |
| | Total dependence in at least one activity (score of 5 or 6) | 17,509 (13.4%) | 7,119 (7.5%) | 2,573 (7.5%) |
| IADL Performance Scale (IALD-H) | Independent or setup help only (score of 0 or 1) | 5,545 (4.2%) | 4,284 (4.5%) | 1,261 (3.8%) |
| | Supervision, limited, or extensive assistance (score of 2 to 4) | 26,697 (20.6%) | 20,158 (21.0%) | 6,539 (19.2%) |
| | Maximal assistance or total dependence (score of 5 or 6) | 99,186 (76.4%) | 72,590 (75.8%) | 26,596 (78.1%) |
| CHESS Score | Any medical instability (score of 1+) | 111,692 (86.0%) | 81,372 (84.9%) | 30,320 (89.1%) |
| Cognitive Performance Scale | Intact (score of 0) | 4,010 (3.1%) | 3,133 (3.3%) | 877 (2.6%) |
| | Borderline intact or mild impairment (score of 1 or 2) | 84,657 (65.2%) | 61,594 (64.3%) | 23,063 (67.7%) |
| | Moderate or moderately severe impairment (score of 3 or 4) | 31,160 (24.0%) | 23,399 (24.4%) | 7,761 (22.8%) |
| | Severe or very severe impairment (score of 5 or 6) | 10,024 (7.7%) | 7,687 (8.0%) | 2,337 (6.9%) |
| **Disease Diagnoses** | | | | |
| Renal Failure | | 9,436 (7.3%) | 6,634 (6.9%) | 2,802 (8.2%) |
| Stroke | | 24,695 (19.0%) | 18,099 (18.9%) | 6,596 (19.4%) |
| Parkinsonism | | 7,275 (5.6%) | 5,314 (5.5%) | 1,961 (5.8%) |
| Cancer | | 12,158 (9.4%) | 8,745 (9.1%) | 3,413 (10.0%) |
| Emphysema/COPD/Asthma | | 18,510 (14.3%) | 13,230 (13.8%) | 5,280 (15.5%) |
| **Behavioural Symptoms (in last 3 days)** | | | | |
| Wandered | | 9,126 (7.0%) | 7,045 (7.4%) | 2,081 (6.1%) |
| Was physically abusive | | 2,828 (2.2%) | 2,191 (2.3%) | 637 (1.9%) |
| Resisted care | | 17,844 (13.7%) | 13,311 (13.9%) | 4,533 (13.3%) |
| **Health Services Utilization** | | | | |
| Number of hospital admissions in last 90 days | Had at least one visit | 56,801 (43.7%) | 41,214 (43.0%) | 15,587 (45.8%) |
| Number of emergency room visits in last 90 days | Had at least one visit | 30,415 (23.4%) | 21,602 (22.5%) | 8,813 (25.9%) |
| **Caregiving/Caregiver Characteristics** | | | | |
| Primary caregiver lives with person | | 66,098 (50.9%) | 48,529 (50.6%) | 17,569 (51.6%) |

(*Continued*)

**Table 1.** (Continued)

| Characteristics of Individuals with Dementia | | n (%) | | |
|---|---|---|---|---|
| | | Total Cohort n = 129,851 | Derivation Cohort n = 95,813 | Validation Cohort n = 34,038 |
| Primary caregiver's relationship with person | Child or child-in-law | 73,066 (56.3%) | 53,746 (56.1%) | 19,320 (56.8%) |
| | Spouse | 38,348 (29.5%) | 28,781 (30.0%) | 9,567 (28.1%) |
| | Other relative | 10,334 (8.0%) | 7,570 (7.9%) | 2,764 (8.1%) |
| | Friend or neighbor | 5,795 (4.5%) | 4,342 (4.5%) | 1,453 (4.3%) |
| | Missing | 2,308 (1.8%) | 1,374 (1.4%) | 934 (2.7%) |
| Primary caregiver is unable to continue | | 26,765 (20.6%) | 18,299 (19.1%) | 8,466 (24.9%) |
| Primary caregiver felt distress, anger, or depression | | 46,011 (35.4%) | 31,804 (33.2%) | 14,207 (41.7%) |
| Hours of informal care | 0 hours | 26,743 (20.6%) | 20,621 (21.5%) | 6,122 (18.0%) |
| | 1 to 24 hours | 67,676 (52.1%) | 49,324 (51.5%) | 18,352 (53.9%) |
| | 25 to 48 hours | 24,698 (19.0%) | 18,053 (18.8%) | 6,645 (19.5%) |
| | Greater than 48 hours | 10,734 (8.3%) | 7,815 (8.2%) | 2,919 (8.6%) |

were mostly a child or child-in-law (56.3%) and co-resided (50.9%). Many caregivers reported being distressed (35.4%) or unable to continue providing care (20.6%).

## Event rates

Within one year of index RAI-HC assessment, 35,513 (37.1%) individuals in the derivation cohort and 10,735 (31.5%) in the validation cohort entered LTC; 11,954 (12.5%) in the derivation cohort and 4,655 (13.7%) in the validation cohort died. The cumulative incidence for LTC entry at 30 days, 90 days, 180 days, and 360 days were 9.95%, 19.2%, 26.5%, and 35.7% respectively. The cumulative incidence curves by risk quintile are shown in Fig 1 (see S1 Fig for the

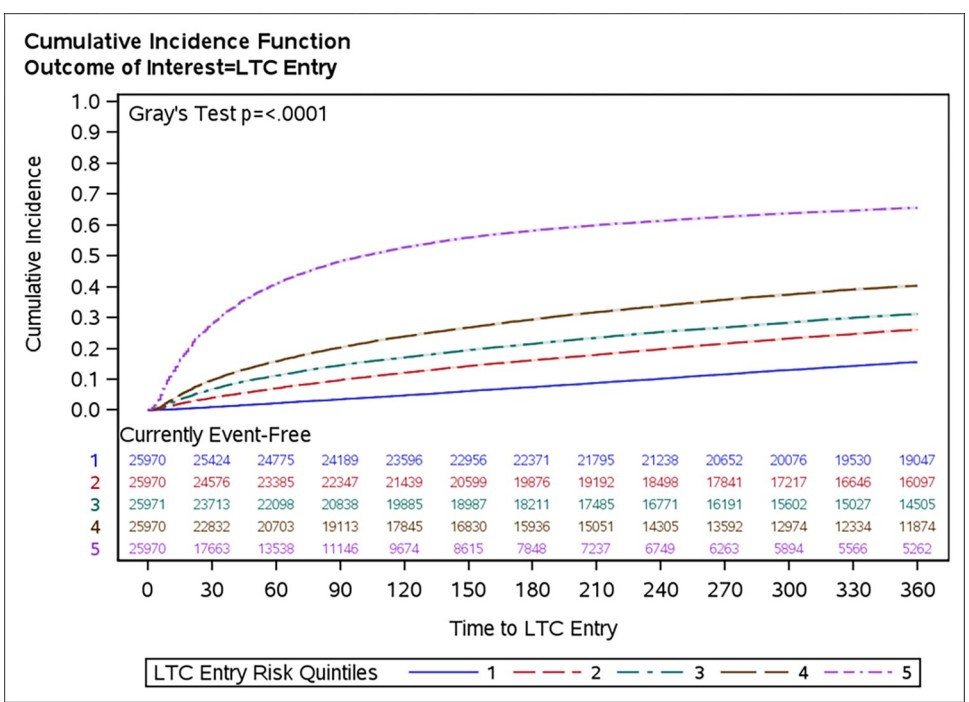

**Fig 1. Cumulative incidence curves for LTC entry, by risk quintile.**

crude cumulative incidence curves for LTC entry and death). Mean and median time to LTC entry were 113 and 78 days, respectively, while mean and median time to death were 135 and 78 days, respectively.

## Model estimates

Subdistribution hazard ratio (sHR) estimates of our model for the total cohort are presented in S2 Table. Individuals' IADL-H score was the most predictive of LTC entry, with sHR increasing from 1.21 (95% CI: 1.01–1.45) for an IADL-H score of 1 to 2.29 (95% CI: 1.97–2.67) for a score of 6 (i.e., total dependence for IADL tasks). Predictors with moderate effects were ADL-H, CPS, and CHESS scores, with higher scores (indicating greater impairment) generally having larger sHRs except for the highest impairment level which either had no effect or reduced the risk of LTC entry. Other factors that moderately increased the risk of LTC entry included having wandering behaviours (sHR = 1.33, 95% CI: 1.28–1.39) and a primary caregiver being unable to continue caregiving (sHR = 1.31, 95% CI: 1.27–1.34). Risk of LTC entry was reduced if patients had university or graduate education (sHR = 0.88, 95% CI: 0.84–0.91), received home health aide or home-making services (sHR = 0.81, 95% CI: 0.78–0.83), co-resided with primary caregiver (sHR = 0.78, 95% CI: 0.75–0.80), and received informal care (sHRs ranging from 0.47 to 0.48).

## Model performance

Our model achieved good discrimination with a c-statistics value of 0.71 for the validation cohort. The model was also well-calibrated, with low $E_{max}$ and $ICI_{avg}$ values of 0.119 and 0.057, respectively. Visual inspection of the calibration plot (Fig 2A) confirms that our model has good calibration overall but over-predicts, especially with increasing risk of LTC entry. Calibration of our algorithm was also good across the various sub-groups examined through our sensitivity analyses (see S2 Fig–S6 Fig for calibration plots).

## Discussion

We developed an algorithm that uses health administrative data representative of the Ontario population to predict time-to-LTC entry among older community-dwelling adults living with dementia. Predictors used in the algorithm included self-reportable health and functional

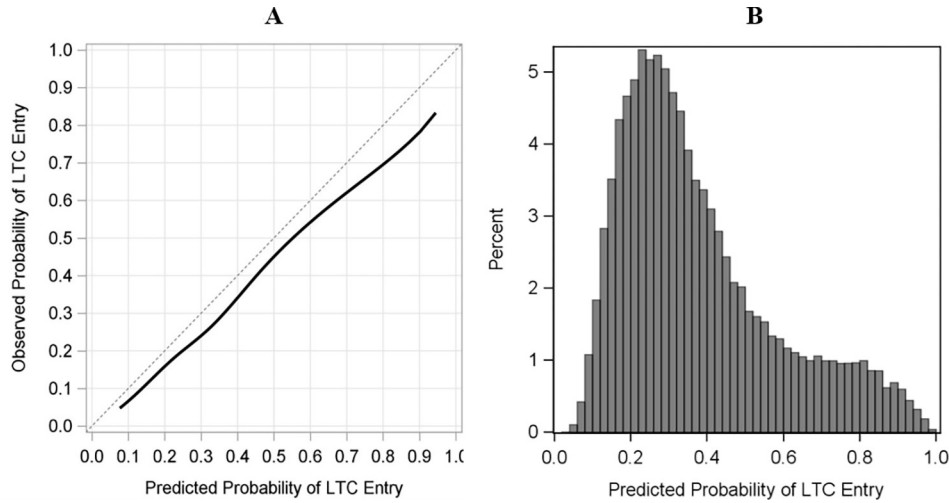

**Fig 2.** A: calibration plot of the observed and predicted probability of LTC entry; sB: distribution of the observed probability of LTC entry.

status, disease diagnoses, and care-related factors. The ability to complete basic and instrumental activities of daily living were most predictive of LTC entry, which concurs with literature on predictors of institutionalization [6,33]. Having a co-residing caregiver reduced the risk of LTC entry while caregiver distress increased the risk, which also align with past findings [33–35] and highlight the importance of accounting for caregiver factors when prognosticating the need for LTC.

Our predictive algorithm has advantages over the MAPLe algorithm that is currently used to prioritize LTC placement in Ontario. Since the MAPLe decision tree uses cognitive impairment in its primary node, 80% of home care clients living with dementia are classified into the two highest MAPLe categories [15], thus limiting its predictive capacity among this cohort. As indicated by our model estimates, many additional patient-level factors, such as comorbid conditions and availability of informal supports, can be used to provide more granular estimates of instantaneous risk of LTC admission. Although these factors are not part of MAPLe, care coordinators do often consider them when assessing an individual's need and urgency for LTC placement. Therefore, while MAPLe remains a useful scale for clinicians, its application as a tool to support patient-oriented risk communication may be limited as individuals with dementia and their families may not be aware of other predictors of LTC entry and do not factor them into care planning. Furthermore, our algorithm provides more precise discrimination for LTC entry, important given the limited number of beds at any given moment. This granularity provides more accurate and fair assessments of who potentially has the greatest need for LTC admission.

Despite the strong performance of our algorithm, the model over-predicted the probability of LTC entry among higher risk individuals; that is, individuals with a higher predicted need for LTC were not placed in LTC within our prediction horizon. This may be partly attributed to our relatively short follow-up period of 1 year–individuals may be correctly identified as at high-risk of LTC entry, but were admitted to LTC right after the 1-year follow-up window. The over-prediction of risk may also be attributed to resource availability, including both home care service capacity, availability of LTC beds, and the number of people on the waitlist for LTC. While the mis-calibration may be correctable by adding health regions as predictors, we chose not to include them to ensure generalizability to other Canadian and international jurisdictions. This observation simultaneously highlights the role of prediction models to inform health system capacity planning in the future and the importance of validating prediction models against variables not included in the algorithm to assess its generalizability.

## Limitations

Despite the large size of our cohort, plethora of predictors, and strong model performance, our study has several limitations. First, we lack information on individuals' and caregivers' preferences for care settings, nor identify which individuals have applied for LTC. Information on preference for care settings is hard to gather on a population-level in most jurisdictions, and preferences also change over time and according to individuals' care needs and caregivers' capacity/burnout [36,37]. Thus, the capturing of preference information in real-time and its incorporation into predictive algorithms is impractical and possibly inappropriate. Secondly, although our algorithm identified caregiver distress as a strong predictor of needing LTC, the complexity of the caregiving experience may not be accurately reflected using broad assessment items. Thirdly, we acknowledge the likely under-capture of dementia cases using health administrative data. While we have improved upon a validated algorithm to capture dementia, administrative data tends to under-capture dementia cases or capture them at a later stage of dementia progression [21,38,39]. Finally, we were restricted to data collected in Ontario and

did not perform external validation. As such, the generalizability of this algorithm to populations in other regions is unknown. In the future, we aim to improve the algorithm performance by exploring the inclusion of system-level factors and performing external validations in other geographic regions.

### Future directions

We have developed this algorithm with the intention to eventually implement it as a web calculator on the Project Big Life (PBL) website (https://www.projectbiglife.ca). This platform has already hosted several of our previously developed algorithms and has garnered millions of views and users around the world. The implementation plans of this algorithm–such as model reduction for parsimony, webpage design, wording and visualization of each question–will be thoroughly discussed and tested with implementation scientists and our patient and caregiver advisors. We will also publish, on the PBL GitHub repository, all associated codes and documentation at the time of launch of this algorithm as a web calculator. These future efforts will ensure that our algorithm will generate wide reach and real-world impact.

## Conclusions and implications

The development of a well-performing prediction model for LTC entry represents the first step towards the creation of an accessible risk communication tool that can support care planning for people living with dementia and their families. This information may be used as the basis for goals-of-care discussions with care providers, ensuring that the transition to LTC best reflects the strengths, preferences and needs of the person living with dementia.

## Supporting information

**S1 Checklist. TRIPOD checklist.**
(DOCX)

**S1 Table. Comprehensive characteristics of the total, derivation, and validation cohorts.**
(DOCX)

**S2 Table. Full regression estimates for the total cohort.**
(DOCX)

**S1 Fig. Cumulative incidence curve for the outcome of long-term care entry and the competing risk of death, for the full cohort.**
(TIF)

**S2 Fig. Calibration plot for the MAPLe model.**
(TIF)

**S3 Fig. Calibration plot stratified by IADL score.**
(TIF)

**S4 Fig. Calibration plot stratified by ADL score.**
(TIF)

**S5 Fig. Calibration plot stratified by CPS score.**
(TIF)

**S6 Fig. Calibration plot by whether individuals were assessed in or out of hospital.**
(TIF)

## Author Contributions

**Conceptualization:** Luke Turcotte, Amy T. Hsu, Peter Tanuseputro, Douglas G. Manuel, Greg Huyer.

**Data curation:** Robert Talarico, Peter Tanuseputro.

**Formal analysis:** Luke Turcotte, Amy T. Hsu, Robert Talarico, Greg Huyer.

**Funding acquisition:** Peter Tanuseputro, Douglas G. Manuel.

**Investigation:** Wenshan Li, Luke Turcotte, Robert Talarico, Greg Huyer.

**Methodology:** Wenshan Li, Luke Turcotte, Amy T. Hsu, Robert Talarico, Greg Huyer.

**Supervision:** Peter Tanuseputro.

**Validation:** Wenshan Li, Robert Talarico.

**Visualization:** Wenshan Li, Robert Talarico.

**Writing – original draft:** Wenshan Li.

**Writing – review & editing:** Wenshan Li, Luke Turcotte, Amy T. Hsu, Robert Talarico, Danial Qureshi, Colleen Webber, Steven Hawken, Peter Tanuseputro, Douglas G. Manuel, Greg Huyer.

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
