## [Decision Letter · Decision Letter 0]

14 Mar 2024

PDIG-D-23-00500

Derivation and Validation of an Algorithm to Predict Transitions from Community to Residential Long-term Care among Persons with Dementia – A Retrospective Cohort Study

PLOS Digital Health

Dear Dr. Li,

Thank you for submitting your manuscript to PLOS Digital Health. After careful consideration, we feel that it has merit but does not fully meet PLOS Digital Health's publication criteria as it currently stands. Therefore, we invite you to submit a revised version of the manuscript that addresses the points raised during the review process.

Please submit your revised manuscript within 60 days May 13 2024 11:59PM. If you will need more time than this to complete your revisions, please reply to this message or contact the journal office at digitalhealth@plos.org. Please include the following items when submitting your revised manuscript:

We look forward to receiving your revised manuscript.

Kind regards,

Ryan S McGinnis

Academic Editor

PLOS Digital Health

Journal Requirements:

1. Please provide separate figure files in .tif or .eps format only and remove any figures embedded in your manuscript file. Please also ensure that all files are under our size limit of 10MB.

2. We do not publish any copyright or trademark symbols that usually accompany proprietary names, eg ©, ®, ™ (e.g. next to drug or reagent names). Please remove all instances of trademark/copyright symbols throughout the text, including ® on page 17.

Additional Editor Comments (if provided):

Thank you for your submission to PLOS Digital Health. The reviewers have provided some thoughtful comments for you to consider in your revision.

Reviewers' comments:

Reviewer's Responses to Questions

**Comments to the Author**

1. Does this manuscript meet PLOS Digital Health’s publication criteria? Is the manuscript technically sound, and do the data support the conclusions? The manuscript must describe methodologically and ethically rigorous research with conclusions that are appropriately drawn based on the data presented.

Reviewer #1: Yes

Reviewer #2: Yes

2. Has the statistical analysis been performed appropriately and rigorously?

Reviewer #1: Yes

Reviewer #2: Yes

3. Have the authors made all data underlying the findings in their manuscript fully available (please refer to the Data Availability Statement at the start of the manuscript PDF file)?

Reviewer #1: No

Reviewer #2: No

4. Is the manuscript presented in an intelligible fashion and written in standard English?

Reviewer #1: Yes

Reviewer #2: Yes

5. Review Comments to the Author

Reviewer #1: This study utilized data from over 100,000 older adults assessed by RAI-HC between 2010 and 2017 to develop and validate a model predicting LTC admissions among individuals with dementia. Both dementia and LTC admission were determined using validated algorithms, and the TRIPOD reporting guideline was followed. The algorithm was developed using self-reportable items to support patient-oriented risk communication. Overall, the study was well-conducted and reported. Here are some minor comments: 

1. The tool is designed to supplement the MAPLe score, an established score for assessing priority levels for LTC admission. The authors stated that 80% of home care clients living with dementia were classified into the two highest MAPLe categories. However, this number was based on a study published in 2014. To more robustly establish the superiority of the currently developed score, the performance of the MAPLe algorithm in the derivation and validation data should be assessed and compared. 

2. It would be helpful for international readers if some background information on the LTC system in Canada could be provided. The authors noted that the data may include individuals who have already applied for LTC. Could you provide information on the median waiting time and how eligibility is determined? Additionally, how might this subset of individuals affect the results, as they are likely to be placed earlier and could potentially inflate accuracy? 

3. The authors note that including CPS > 2 was shown internally to increase the number of dementia cases captured by 8%. Is there a reference available for this? Additionally, please confirm whether it is CPS = 2 or CPS ≥ 2. 

4. Age was modeled using restricted cubic splines with 5 knots. Could you provide the rationale for not including age in years in its original format? 

5. How was collinearity defined in this study? Was there a pre-specified cutoff value used?

Reviewer #2: Using a number of administrative data sources in Canada, the authors develop and validate a prediction model for transitioning to long-term care amongst older adults with dementia. Given that the disease course for adults with dementia is highly variable, often with significant resource implications, there is a clear need for prognostic tools both for patients/caregivers and providers/health systems. I think the work here is also well-motivated by highlighting the deficiencies of the MAPLe tool that is currently being used in Canada. Statistically, the authors correctly employ an approach that accounts for the competing risk of death, evaluate both discrimination and calibration, and while there is a not a true external validation, the design of using a past dataset to predict the future is how such models would be deployed in practice. Overall, I think the predictive performance of the model is about what I’d expect, good, but not great. That’s less of an issue of the data available or modeling approach, but just reflects the reality that predicting shorter term outcomes in these types of populations is challenging because a component of what occurs is inevitably random/stochastic and not predictable. Below I offer some suggestions for improvement related to potential comparisons. 

Major Comments

1. I think the paper would strengthened by evaluating the performance of the predictive model for 1-year mortality of Deardorff et al. (JAMA Intern Med. 2022 Nov 1;182(11):1161-1170. doi: 10.1001/jamainternmed.2022.4326.) in the validation cohort. While that is designed for a different outcome in individuals with dementia, there is a natural question of whether we need yet another model, or perhaps those at high mortality risk are also generally the same individuals at high risk of transitioning to long-term care? A risk equation derived specifically for long-term care transitions should perform better, but the obvious question is by how much? 

2. The comparison to Deardorff et al. also raises the issue of model availability. Their model is available as a web calculator on ePrognosis.org. As is stands, while someone could get most of the way if they wanted to evaluate this model in a different cohort, there are issues with re-creating the cubic splines with age and not having a way to access the non-parametrically estimated baseline component of the Fine-Gray model. It would be ideal to provide a R model object that could be used for prediction, perhaps in tandem with a R/Shiny web calculator. I do recognize that both of those outlets implicitly need to have the underlying dataset available/stored, which is probably an issue given the provided statement on data sharing from ICES. There may not be a solution here, but I strongly believe its important to promote risk stratification tools that can be openly evaluated, locally calibrated, etc. 

3. The RAI-HC assessment provides an impressive wealth of data, particularly around functional status which is often not available in claims or the electronic health record. But, a natural question is whether you need all the variables included in the current model, or would a more parsimonious model perform about the same? This is an important issue because in the US, for example, I doubt we’d ever have this breadth of information available clinically. I think a way to approach this would be to compare to a simplified model that mimics what would be available in the electronic health record. So, something like, don’t know education and marital status, or place of residence, have ADL/IADL but not the other assessments, and then just have information on comorbidity and medications? 

Minor Comments

1. Abstract. I would add some result that quantitatively reflects calibration to the results section.

2. I believe information on race or ethnicity is not collected in the data assembled at ICES, correct? Presumbly this reflects a large majority white population, so it is unknown how this model would perform in more diverse populations, whether the model is fair, etc. I would add a thought along these lines to the limitations. 

3. Another limitation that should be acknowledged is that the whole analysis rests on a dementia ascertainment mechanism that almost certainly underestimates the community-burden of dementia, and likely concentrates on those at later stages of the disease. I think the algorithm employed is as good as one can do from administrative data, but there is no way to overcome the reality that dementia tends to be under-coded, and when it is, it is later stages of the disease. Could cite work such as Tsoy et al. (JAMA Neurol. 2021;78(6):657–665. doi:10.1001/jamaneurol.2021.0399) that generally demonstrate this issue. 

4. Lines 117-118. I would add a brief sentence here that summarizes what variables tended to be missing the most along with frequency. 

5. Lines 129-130. “Finally, model coefficients for algorithm implementation were generated based on the total cohort.” 

6. Line 162. “Subdistribution hazard ratio (sHR) estimates….” I typically use this labeling of sHR to differentiate that you’re using Fine-Gray and not a cause-specific Cox model. 

7. I would add the histogram of the predicted risk distribution in Figure S2 to the calibration plot shown in Figure 2.

6. PLOS authors have the option to publish the peer review history of their article (what does this mean?). If published, this will include your full peer review and any attached files.

**Do you want your identity to be public for this peer review?** For information about this choice, including consent withdrawal, please see our Privacy Policy.

Reviewer #1: No

Reviewer #2: Yes: Nicholas M. Pajewski

---

## [Decision Letter · Decision Letter 1]

10 Jul 2024

PDIG-D-23-00500R1

Derivation and Validation of an Algorithm to Predict Transitions from Community to Residential Long-term Care among Persons with Dementia – A Retrospective Cohort Study

PLOS Digital Health

Dear Dr. Li,

Thank you for submitting your manuscript to PLOS Digital Health. After careful consideration, we feel that it has merit but does not fully meet PLOS Digital Health's publication criteria as it currently stands. Therefore, we invite you to submit a revised version of the manuscript that addresses the points raised during the review process.

Please submit your revised manuscript within 30 days Aug 09 2024 11:59PM. If you will need more time than this to complete your revisions, please reply to this message or contact the journal office at digitalhealth@plos.org. Please include the following items when submitting your revised manuscript:

We look forward to receiving your revised manuscript.

Kind regards,

Ryan S McGinnis

Academic Editor

PLOS Digital Health

Journal Requirements:

Additional Editor Comments (if provided):

Reviewers' comments:

Reviewer's Responses to Questions

**Comments to the Author**

1. If the authors have adequately addressed your comments raised in a previous round of review and you feel that this manuscript is now acceptable for publication, you may indicate that here to bypass the “Comments to the Author” section, enter your conflict of interest statement in the “Confidential to Editor” section, and submit your "Accept" recommendation.

Reviewer #1: All comments have been addressed

Reviewer #2: (No Response)

2. Does this manuscript meet PLOS Digital Health’s publication criteria? Is the manuscript technically sound, and do the data support the conclusions? The manuscript must describe methodologically and ethically rigorous research with conclusions that are appropriately drawn based on the data presented.

Reviewer #1: Yes

Reviewer #2: Yes

3. Has the statistical analysis been performed appropriately and rigorously?

Reviewer #1: Yes

Reviewer #2: Yes

4. Have the authors made all data underlying the findings in their manuscript fully available (please refer to the Data Availability Statement at the start of the manuscript PDF file)?

Reviewer #1: No

Reviewer #2: No

5. Is the manuscript presented in an intelligible fashion and written in standard English?

Reviewer #1: Yes

Reviewer #2: Yes

6. Review Comments to the Author

Reviewer #1: The authors have successfully addressed my comments.

Reviewer #2: I appreciate the author’s thoughtful edits and responses to my initial critique (Reviewer #2). I have only a few minor additional suggestions. 

With respect to model dissemination, I think it would be help to add a brief paragraph, perhaps titled “Future Directions”, right before the Conclusion. I would add much of what was said in response to my initial bullet # 2. Even if this model is not immediately available now, I think it’s important to point readers to the PBL website, and to mention your intention to pursue further external validation and also pursuing user engagement before deploying this model. Don’t need a lengthy discussion, but this will help a reader know where to look for future progress. 

With respect to my minor comment #2, I’m curious if the provided population statistics are restricted to older populations? For example, in the US, while there are ever growing populations of Blacks, LatinX, Asians, etc., the population is still very much majority white in the age range where dementia tends to occur (roughly 70 or 75 years plus). I’m going to guess it’s similar in Ontario. I do think a qualifier needs to be added to the limitations that mentions that generalizability to broader populations is unknown, and is an important topic for future development of the algorithm. 

Abstract, Conclusions and Implications. Perhaps rephrase as “We developed an algorithm to predict to time LTC entry among individuals living dementia that improves upon an existing tool used for prioritization.” Well-performing is a bit of a nebulous term here. 

Lines 213-222. One other thought that you could mention is that the calibration issue may also reflect artificially ending the time horizon at 1 year. If many of these individuals ended up getting placed in LTC in say the following year, the model isn’t “wrong” for assigning higher risk to these individuals. Certainly see this type of pattern in trying to predict mortality, where short term outcomes are very, very hard to calibrate, but as one moves out to 2 or 3-year mortality, outcomes tend to become much more predictable and one gets better calibration.

7. PLOS authors have the option to publish the peer review history of their article (what does this mean?). If published, this will include your full peer review and any attached files.

**Do you want your identity to be public for this peer review?** For information about this choice, including consent withdrawal, please see our Privacy Policy. 

Reviewer #1: No

Reviewer #2: Yes: Nicholas M. Pajewski

---

## [Decision Letter · Decision Letter 2]

14 Aug 2024

Derivation and Validation of an Algorithm to Predict Transitions from Community to Residential Long-term Care among Persons with Dementia – A Retrospective Cohort Study

PDIG-D-23-00500R2

Dear Dr Li,

We are pleased to inform you that your manuscript 'Derivation and Validation of an Algorithm to Predict Transitions from Community to Residential Long-term Care among Persons with Dementia – A Retrospective Cohort Study' has been provisionally accepted for publication in PLOS Digital Health.

Best regards,

Ryan S McGinnis

Academic Editor

PLOS Digital Health

Reviewer Comments (if any, and for reference):

Reviewer's Responses to Questions

**Comments to the Author**

1. If the authors have adequately addressed your comments raised in a previous round of review and you feel that this manuscript is now acceptable for publication, you may indicate that here to bypass the “Comments to the Author” section, enter your conflict of interest statement in the “Confidential to Editor” section, and submit your "Accept" recommendation.

Reviewer #2: All comments have been addressed

2. Does this manuscript meet PLOS Digital Health’s publication criteria? Is the manuscript technically sound, and do the data support the conclusions? The manuscript must describe methodologically and ethically rigorous research with conclusions that are appropriately drawn based on the data presented.

Reviewer #2: Yes

3. Has the statistical analysis been performed appropriately and rigorously?

Reviewer #2: Yes

4. Have the authors made all data underlying the findings in their manuscript fully available (please refer to the Data Availability Statement at the start of the manuscript PDF file)?

Reviewer #2: No

5. Is the manuscript presented in an intelligible fashion and written in standard English?

Reviewer #2: Yes

6. Review Comments to the Author

Reviewer #2: The authors have addressed the handful of minor issues I raised in my prior review.

7. PLOS authors have the option to publish the peer review history of their article (what does this mean?). If published, this will include your full peer review and any attached files.

**Do you want your identity to be public for this peer review?** For information about this choice, including consent withdrawal, please see our Privacy Policy.

Reviewer #2: **Yes: **Nicholas M. Pajewski
